# Performance of Hypersaline Brine Desalination Using Spiral Wound Membrane: A Parametric Study

**DOI:** 10.3390/membranes13020248

**Published:** 2023-02-19

**Authors:** Kathleen Foo, Yong Yeow Liang, Woei Jye Lau, Md Maksudur Rahman Khan, Abdul Latif Ahmad

**Affiliations:** 1Faculty of Chemical and Process Engineering Technology, Universiti Malaysia Pahang, Lebuh Persiaran Tun Khalil Yaakob, Kuantan 26300, Malaysia; 2Advanced Membrane Technology Research Centre (AMTEC), Faculty of Chemical and Energy Engineering, Universiti Teknologi Malaysia, Johor Bahru 81310, Malaysia; 3Petroleum and Chemical Engineering Programme Area, Faculty of Engineering, Universiti Teknologi Brunei, Gadong BE1410, Brunei; 4School of Chemical Engineering, Engineering Campus, Universiti Sains Malaysia, Nibong Tebal 14300, Malaysia

**Keywords:** module-scale analysis, hypersaline brine desalination, reverse osmosis, specific energy consumption, concentration polarization

## Abstract

Desalination of hypersaline brine is known as one of the methods to cope with the rising global concern on brine disposal in high-salinity water treatment. However, the main problem of hypersaline brine desalination is the high energy usage resulting from the high operating pressure. In this work, we carried out a parametric analysis on a spiral wound membrane (SWM) module to predict the performance of hypersaline brine desalination, in terms of mass transfer and specific energy consumption (SEC). Our analysis shows that at a low inlet pressure of 65 bar, a significantly higher SEC is observed for high feed concentration of brine water compared with seawater (i.e., 0.08 vs. 0.035) due to the very low process recovery ratio (i.e., 1%). Hence, an inlet pressure of at least 75 bar is recommended to minimise energy consumption. A higher feed velocity is also preferred due to its larger productivity when compared with a slightly higher energy requirement. This study found that the SEC reduction is greatly affected by the pressure recovery and the pump efficiencies for brine desalination using SWM, and employing them with high efficiencies (*η_R_* ≥ 95% and *η_pump_* ≥ 50%) can reduce SEC by at least 33% while showing a comparable SEC with SWRO desalination (<5.5 kWh/m^3^).

## 1. Introduction

The disposal of hypersaline brine has been a subject of rising environmental concern. This is because an inappropriate handling of brine discharged from seawater/brackish water desalination plants into the environment would inflict a tremendous amount of damage to the surrounding ecosystem as it could disrupt the maritime concentration balance and contaminate the soil [1,2,3,4]. Hence, there is a growing interest in recent years focusing on the coupling of desalination plants with brine management to utilise the high-salinity water while reducing the volume of liquid waste. The ultimate goal of the brine management is to achieve a zero liquid discharge (ZLD) [4,5,6]. When water is removed from the high-salinity feed and only solids are left as the final waste product, this management could reduce the harmful impact of hypersaline brine as well as minimise the risks of contamination on the environment due to the improper disposal of brines and solid waste [5,6,7,8,9].

Desalination of brine water can be carried out using thermal-based or membrane-based methods [1,4,10]. One of the most important thermal-based technologies for brine water treatment is multi-stage flash distillation (MSF) which accounts for about 64% used in thermal desalination [11]. A major challenge for the commonly used thermal-based method is its extremely high energy consumption (i.e., 52–70 kWh/m^3^) involved in evaporating the high-salinity water and reducing the brine discharge [4,10]. The membrane-based method, on the other hand, has a significantly lower energy consumption of 7–12 kWh/m^3^ for brine desalination in reverse osmosis (RO) processes [12], despite the fact that RO is still an energy-intensive process for drinking water production.

Nonetheless, it must be pointed out that RO process does not require a phase change of feed stream, which explains the growing interest of the membrane-based method for brine desalination. In order to desalinate the high-salinity brine, the RO membrane module would have to operate at a much higher pressure condition to separate water from the highly concentrated solution [13]. One of the latest approaches to improve the efficiency of hypersaline brine desalination is by employing osmotically assisted reverse osmosis (OARO), which combines both forward osmosis (FO) and RO principles. In OARO, a less concentrated solution is used in the permeate side of RO compared with the feed solution in order to reduce the osmotic pressure differences across the membrane. Thus, a smaller hydraulic pressure can be applied compared with traditional RO which not only allows for the treatment of a high salinity solution but also avoids exceeding the burst pressure of the membrane module [14]. Feed temperature is acknowledged as an important parameter of the design and operation of the RO system. For instance, it was found by Goosen et al. [15] that doubling the feed temperature from 20 °C to 40 °C increases the water flux by 60%. However, the positive effects of increasing feed temperature were found to be more significant for low salinity feed water, compared with high salinity water [16].

Nowadays, the most common membrane module used in desalination is the spiral wound membrane (SWM) module [17] which, to date, can withstand pressure of up to 80 bar [18,19]. The use of higher pressure beyond the practical limit of the conventional SWM module is not possible for the current state-of-the-art techniques due to higher risks of membrane damage and leaking issues associated with the membrane module [20,21,22]. Nevertheless, several studies have reported that this problem could be solved by using high-strength adhesives for gluing the membrane sheets together [21,23] and/or additional sealing techniques of membrane sheets [24,25] to improve module durability. Not only that, the development of high-strength materials and/or more robust module designs would also help to overcome the pressure limit of the conventional SWM module in the future, owing to the extensive progress in current membrane research [4,12,26,27,28,29,30]. Based on this, it is believed that the use of the SWM module with a higher pressure for hypersaline brine desalination may be a reality in the years to come.

In spite of this, no parametric analysis has been carried out to investigate the impact of typical hypersaline brine condition using SWM on the energy consumption. Despite several studies using SWM for predicting the energy consumption, they are restricted to the typical RO feed condition [31,32]. While the traditional high rejection seawater reverse osmosis (SWRO) has a typical water flux of 30–40 L/m^2^.h using the commercial SWM module (note: membranes were tested based on standard seawater conditions of 32,000 ppm NaCl at 55 bar) [24,33,34,35], the flux performance of hypersaline brine reverse osmosis (HBRO) using the SWM module remains largely unclear.

Hence, this study aims to perform a parametric analysis on hypersaline brine desalination using the module-based model and compare its performance against the conventional SWRO for a full-scale SWM module, in terms of permeate flux, concentration polarisation and energy consumption. The effects of feed conditions, such as inlet operating pressure, feed concentration and inlet velocity on HBRO performance are also evaluated. Given that the energy requirement of brine desalination (7 to 12 kWh/m^3^ [12]) is significantly higher than those for using SWRO (1.5 to 5.5 kWh/m^3^ [36]), this means that a substantial reduction of energy for brine water is needed in order to ensure it is practical to be used. For this reason, energy recovery parameters, such as the retentate pressure recovery efficiency (*η_R_*) and pump efficiency (*η_pump_*) are employed as means to reduce energy. Moreover, both parameters have yet to be tested for brine desalination. This is because these parameters directly affect the amount of energy conserved during the pressure recovery process [37], which are beneficial for minimising the energy consumption of the membrane operation. Not only that, it is also important to gain understanding of the degree of energy reduction that can be achieved with respect to the energy recovery parameters used in the process.

## 2. Methodology

### 2.1. Module-Scale Analysis

Figure 1 shows the schematic diagram of the typical RO membrane desalination process used for the module-scale analysis. The process is made up of four single-stage module units with each unit consisting of 1 m of membrane length [32,37]. The feed solution, i.e., seawater or hypersaline brine, is pressurised before entering the RO membrane module to produce high-purity water (i.e., permeate), while the rejected solution (i.e., concentrate) is sent through a pressure recovery unit to recover some of the solution pressure. Further, the module is assumed to have a typical ladder-type commercial feed spacers [37].

A multi-scale approach is used in this study in which the correlations for the Sherwood number and friction factor from a small-scale computational fluid dynamic (CFD) analysis are employed to estimate the specific energy consumption (SEC) of the SWM performance. The capability of this method has been proven in the literature [32,38], as it could interplay not only the permeate flux, but also the concentration polarisation (CP) as well as the pressure drop of the entire SWM. The module-scale model solves a series of one-dimensional ordinary differential equations (ODEs) to calculate for the pressure drop and global and salt mass balances along a full-length membrane module, as expressed in the following equations:(1)$$\frac{dQ}{dx}=-\frac{A_{m}J}{\rho L_{m}}$$
(2)$$\frac{dw_{b}}{dx}=\frac{A_{m}J}{\rho QL_{m}}\left(w_{b}-w_{p}\right)$$
(3)$$\frac{dp}{dx}=\frac{2\rho u_{eff}^{2}}{d_{h}}f_{glob}$$
where *Q* is the volumetric flow rate, *A_m_* is the membrane area, *J* is the permeate flux, *ρ* is the density, *L_m_* is the membrane length, *p* is the pressure, *u_eff_* is the effective velocity, *d_h_* is the hydraulic diameter, *f_glob_* is the Fanning friction factor, *w_b_* and *w_p_* are the solute mass fraction of feed and permeate, respectively. The permeate flux (*J*) is calculated based on the well-known Kedem–Katchalsky model [39]:(4)$$J=L_{p}\left(\Delta p_{tm}-\sigma \varphi Rw_{w}\right)$$
where *L_p_* is the membrane permeance, Δ*p_tm_* is the transmembrane pressure, *σ* is the reflection coefficient, *φ* is the osmotic pressure coefficient, *R* is the membrane intrinsic rejection and *w_w_* is the solute mass fraction near the membrane boundary layer.

The total volumetric permeate flow rate (*Q_p_*) can then be calculated by integrating the flux along the membrane module:(5)$$Q_{p}=\delta_{ch}\int_{0}^{L_{m}}\frac{J}{\rho }dx$$
where *δ_ch_* is the channel width of the membrane module. Further details on the flux calculation can be found elsewhere [37,40] and the solute concentration at the membrane wall (*w_w_*) can be calculated using the equation, as follows [40]:(6)$$w_{w}=w_{p}+\frac{1}{2\sigma \varphi }\left(\Delta p_{tm}-\frac{k_{mt}}{L_{p}}\right)+\sqrt{{\left[\frac{1}{2\sigma \varphi }\left(\Delta p_{tm}-\frac{k_{mt}}{L_{p}}\right)\right]}^{2}+\frac{k_{mt}}{\sigma \varphi L_{p}}\left(w_{b}-w_{p}\right)}$$
where *k_mt_* is the mass transfer coefficient.

Equations (1) to (3) are derived from the mass and momentum balances, which can be solved by the Runge–Kutta method. The mass transfer, as well as the pressure drop are calculated based on the correlations for the dependencies of the Sherwood number (*Sh)* and *f_glob_* on the Reynolds number (*Re_h_*), as expressed in Equations (7) and (8), respectively. Both equations are developed based on the validated small-scale CFD analysis for the typical RO unit-scale model using a ladder-type spacer [32,37]:(7)Sh=kmtdhD=2.44Reh0.61
(8)fglob=dh2ρueff2ΔpchLm=8.76Reh−0.62
where *Sh* represents the dimensionless approximation for the mass transfer performance, while the *f_glob_* is the proxy measure of pressure loss across the membrane channel.

It is important to note that the mass transfer correlation in Equations (7) is determined considering the impermeable wall condition [37]. Therefore, the mass transfer data must be converted to the case of the permeable wall condition when performing the module-scale analysis. This can be carried out given that the ratio of the volumetric flux to the impermeable mass transfer coefficient (*ψ*) is not more than 20 [41]. In this case, *ψ* is below 2 for all case studies considered, hence the correlations can be applied to predict the mass transfer coefficient (*k_mt,per_*) under the permeable wall condition [41,42,43,44,45,46].

Note that the module-scale analysis is performed considering only the fluid flow in the membrane feed side of the SWM module, as the pressure drop in the permeate side is often considered negligible due to the very high transmembrane pressure, when compared with the permeate pressure [32,37]. Furthermore, the CP effect could be neglected at the membrane permeate side because of the high intrinsic rejection of the RO membrane. Hence, it is safe to model only the module feed side.

### 2.2. Analysis of Results

A CP modulus (γ) is given, as in Equation (9), to predict the extent of concentration polarisation in the membrane module. In this case, a larger value of γ indicates a higher CP in the membrane channel and vice versa [40,47]:(9)$$\gamma =\frac{w_{w}-w_{p}}{w_{b}-w_{p}}$$

The area-averaged values of the local variables (*ϕ*), such as the permeate flux and CP modulus along the membrane module length (*L_m_*) can be calculated, as follows [32,48]:(10)$$\bar{\phi }=\frac{1}{L_{m}}\int \phi dx$$

The performance metric with respect to SEC is calculated based on the total permeate flow rate and pressure drop obtained from the module-scale model as expressed in Equation (11) [49]:(11)$$\mathrm{SEC}=\frac{\left[\Delta p_{tm}-\eta_{R}\left(\Delta p_{tm}-\Delta p_{ch}\right)\left(1-R_{RO}\right)\right]}{R_{RO}\eta_{pump}}$$
where Δ*p_tm_* is the transmembrane pressure $\left(\Delta p_{tm}=p_{in}-p_{a}\right)$, *η_R_* is the retentate pressure recovery efficiency, Δ*p_ch_* is the pressure drop in the membrane module, *R_RO_* is the recovery ratio of the RO process and *η_pump_* is the pump efficiency. The recovery ratio (*R_RO_*) is defined as the ratio of the total permeate flow rate to the feed flow rate (*Q_p_*/*Q_f_*), while *η_R_* refers to the efficiency of the pressure recovery unit (as in Figure 1) for recovering energy from the retentate pressure exiting from the RO membrane module [37,44]. SEC is widely used to measure the energy requirement of any desalination system and it is generally referred to as the ratio of energy consumed in a RO membrane module to the unit volume of permeate water generated [50,51,52,53]. The details for the derivation of SEC can be found elsewhere [49].

### 2.3. Comparison between HBRO and SWRO

Table 1 describes the case parameters used for SWRO and HBRO, respectively. A minimum solute feed mass fraction of 0.05 is considered for hypersaline brine (HB) as the high-salinity water has an average total dissolved solid (TDS) greater than 35,000 mg/L of typical seawater [12]. The inlet operating pressure, *p_in_*, on the other hand, is varied in the range of 65–80 bar for both SW and HB desalination (Table 1) as the upper limit (80 bar) is the current practical limitation of RO membranes [18,19]. In terms of feed velocities, the range of values selected (as in Table 1) corresponds to a hydraulic *Re_h_* of not more than 250 [40,54], which is within the flow velocities used in the RO membranes for the typical channel height considered [32]. The dimensional specifications of the SWM RO module, as well as the base conditions used for the case analysis are summarised in Table 2.

## 3. Results and Discussion

### 3.1. Effect of the Feed Conditions on HBRO Performance

Figure 2 compares the performance of HBRO with conventional SWRO at different inlet operating pressures (*p_in_*) by means of the permeate flux, concentration polarisation, mass transfer coefficient and SEC. It is worth noting that the module-based model is developed based on the validated small-scale CFD analysis of the SWM desalination process using a conventional feed spacer [32,37]. The mass transfer coefficient $\left({\bar{k}}_{mt}\right)$ calculated in this study depends not only on the feed condition and properties, but also on the feed spacer geometry. In our simulation study, the mass transfer coefficients calculated (i.e., 6 × 10^−5^ to 8 × 10^−5^ m/s) agree reasonably with the value obtained in the literature (i.e., 6.61 × 10^−5^ m/s [37]) at a similar feed condition for dual-layer feed spacer geometry. Lastly, most of the SEC values calculated for seawater and hypersaline brine desalination in this study are below 3 kWh/m^3^ and 6 kWh/m^3^, respectively (see Figure 2 and Figure 3), which agree reasonably with the data reported in the literature (1.5 to 5.5 kWh/m^3^ for seawater [55] and 7 to 12 kWh/m^3^ for brine water [12]), thus providing confidence in the prediction of ${\bar{k}}_{mt}$ and SEC in this case.

Compared with the permeate flux obtained for SWRO in the range of 30–40 L/m^2^.h (as in Figure 2a), HBRO demonstrates a permeate flux of no more than 28 L/m^2^.h due to a much higher solute concentration of the brine water than seawater. Interestingly, at a low inlet pressure of 65 bar, Figure 2d shows a very high energy requirement of up to 21 kWh/m^3^ in the case of HBRO for a feed solute mass fraction of 0.08, which is about 2.5 times compared with the SWRO for the feed solute mass fraction of 0.035. This is because, under this condition, the recovery ratio (*R_RO_*) obtained is very low (i.e., 1%) to desalinate a very high concentration of brine water (i.e., *w_b_*_0,HB_ = 0.08) with a small inlet pressure. This implies that the inlet pressure should be at least 75 bar for the desalination of highly concentrated brine water, in order to significantly reduce the energy consumption. Note that pressure has a direct relationship to SEC while the productivity has a reverse relationship with SEC.

Figure 3 discusses the effect of the inlet velocity (*u_avg_*) on the overall performance of hypersaline brine desalination in an SWM module. The range of inlet velocity used in this study is set within 0.135 m/s, as recommended by the SWM manufacturers [56]. Thus, the impact due to an upper limit of feed velocity considered in this work on the membrane is negligible in this case. Figure 3a shows that an increase in the inlet velocity of the feed brine results in a higher flux due to enhanced mass transfer (Figure 3b) [57], for all inlet pressures studied in this work. In terms of energy consumption, the SEC only increases by less than 7.5% as the inlet velocity increases (Figure 3c). This is because, under a constant inlet pressure, an increase in the inlet velocity tends to decrease the recovery ratio (see Figure 3d), leading to a higher SEC. Given that there is only a small increase in SEC (i.e., <7.5%) due to the increase in inlet velocity, it is preferable for hypersaline brine desalination to operate at a high inlet velocity under a constant inlet pressure condition.

### 3.2. Effect of Retentate Pressure Recovery and Pump Efficiency

This section studied the impact of retentate pressure recovery and pump efficiencies on the performance of hypersaline brine desalination using SWM. The upper limit of retentate pressure recovery efficiency used in this study is 95% given that the current pressure recovery device can achieve an efficiency of over 95% [58,59], whereas the pump efficiency could reach a practical limit of about 90% if the centrifugal pump type is used [58,60]. The improvement in the energy recovery efficiencies can be achieved by increasing the processing capacity of the pressure recovery device [61], or using a larger size of pump [58,59]. A higher energy recovery efficiency typically reduces the energy consumption as more energy can be recovered [58,59,60]. Figure 4 shows that an increase in the retentate pressure recovery efficiency (*η_R_*) and/or pump efficiency (*η_pump_*) would reduce the SEC of HBRO by at least 33%. These results indicate that a high retentate pressure recovery efficiency is very important for SEC reduction, especially at a low inlet pressure condition (i.e., 65 bar) as a maximum 77% of energy saving can be achieved (Figure 4a). This study found that all of the inlet pressures used for hypersaline brine desalination show a comparable SEC with SWRO desalination (<5.5 kWh/m^3^ [36]) at *η_R_* of 95% (Figure 4a) and/or a minimum *η_pump_* of 50% (Figure 4b). However, a high energy requirement (i.e., >8 kWh/m^3^) is required for *η_R_* of 50%, despite the fact that a high efficiency pump (i.e., *η_pump_* = 85%) is used (see Figure 4a). Fouling remains the bottleneck, impeding the understanding of its effect on the flux and energy consumption using SWM. Thus, future research is still needed to enable comprehensive understanding of fouling on energy consumption, especially for typical hypersaline feed conditions.

## 4. Conclusions

A parametric analysis is performed in this study to evaluate the overall performance of hypersaline brine desalination in an SWM module using a module-scale model. At a lower range of inlet pressure (i.e., 65 bar), it is interesting to note that when the solute concentration of brine water is more than twice the solute concentration of seawater (i.e., 0.08 vs. 0.035), a significantly higher SEC is observed due to a very low recovery rate (i.e., 1%). Our findings indicate that the inlet pressure should be at least 75 bar for the desalination of highly concentrated brine water in order to minimise the energy consumption. This study also found that a higher feed velocity is preferred due to its larger productivity while only requiring a slightly higher energy requirement. The analysis also reported that the SEC reduction is greatly affected by the pressure recovery and pump efficiencies in desalinating brine water using SWM, and employing them with high efficiency (*η_R_* ≥ 95% and *η_pump_* ≥ 50%) can reduce SEC by energy-savings of at least 33% while showing a comparable SEC with SWRO desalination (<5.5 kWh/m^3^).

## Figures and Tables

**Figure 1 membranes-13-00248-f001:**
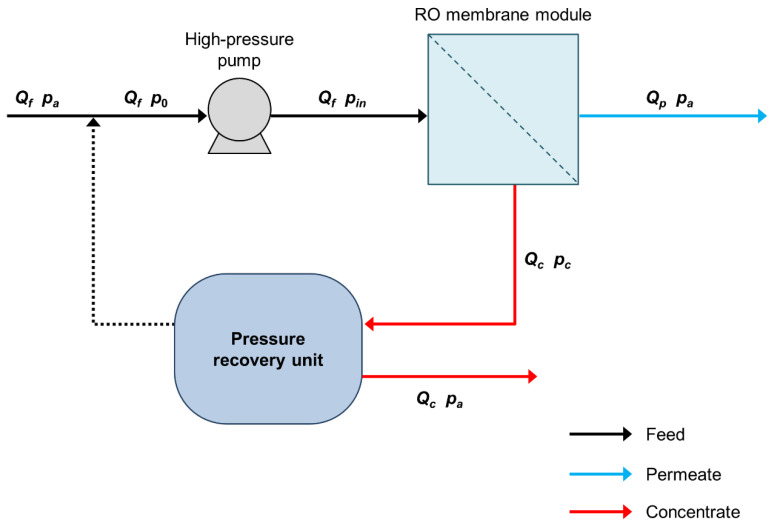
Schematic illustration of the typical RO membrane desalination process.

**Figure 2 membranes-13-00248-f002:**
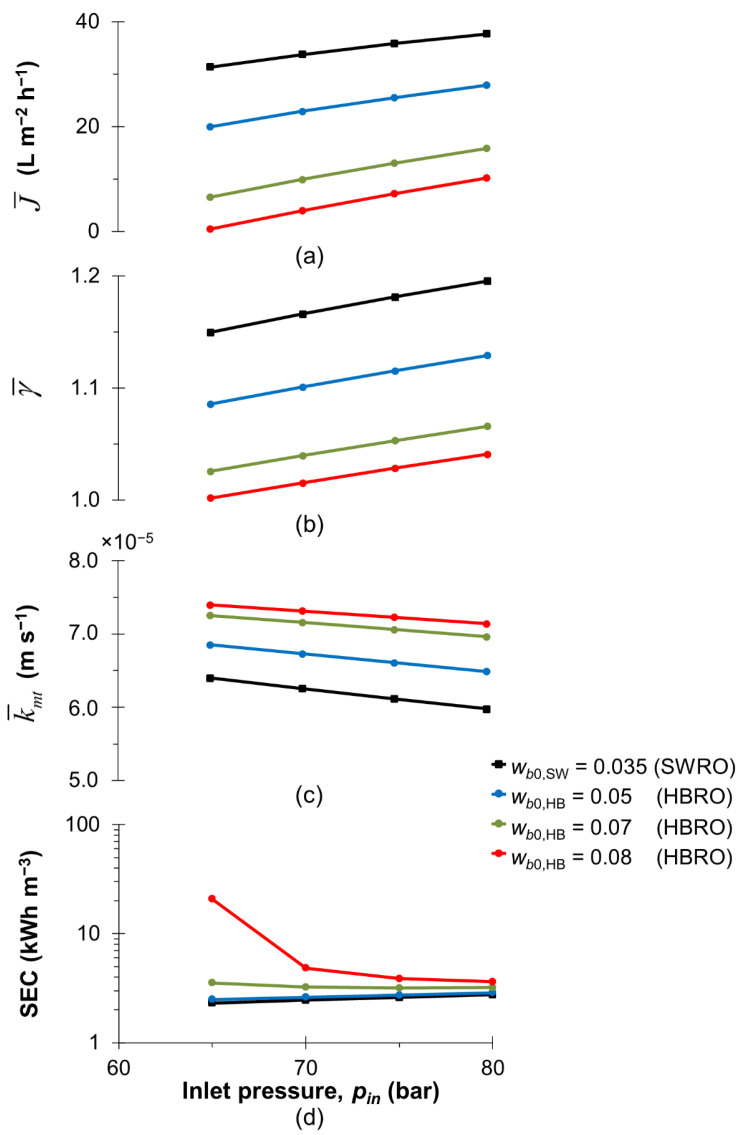
Comparison between SWRO and HBRO performance in terms of (**a**) the area-averaged flux $\left(\bar{J}\right)$, (**b**) area-averaged concentration polarisation modulus $\left(\bar{\gamma }\right)$, (**c**) area-averaged mass transfer coefficient $\left({\bar{k}}_{mt}\right)$ and (**d**) specific energy consumption at a different inlet pressure (*p_in_*) and feed mass fraction (*w_b_*_0_). *w_b_*_0,SW_ and *w_b_*_0,HB_ represent the feed mass fraction of seawater and hypersaline brine, respectively. The comparison study is performed at the same inlet velocity, i.e., *u_avg_* = 0.135 m s^−1^, with *η_R_* and *η_pump_* constant at 0.95 and 0.85, respectively.

**Figure 3 membranes-13-00248-f003:**
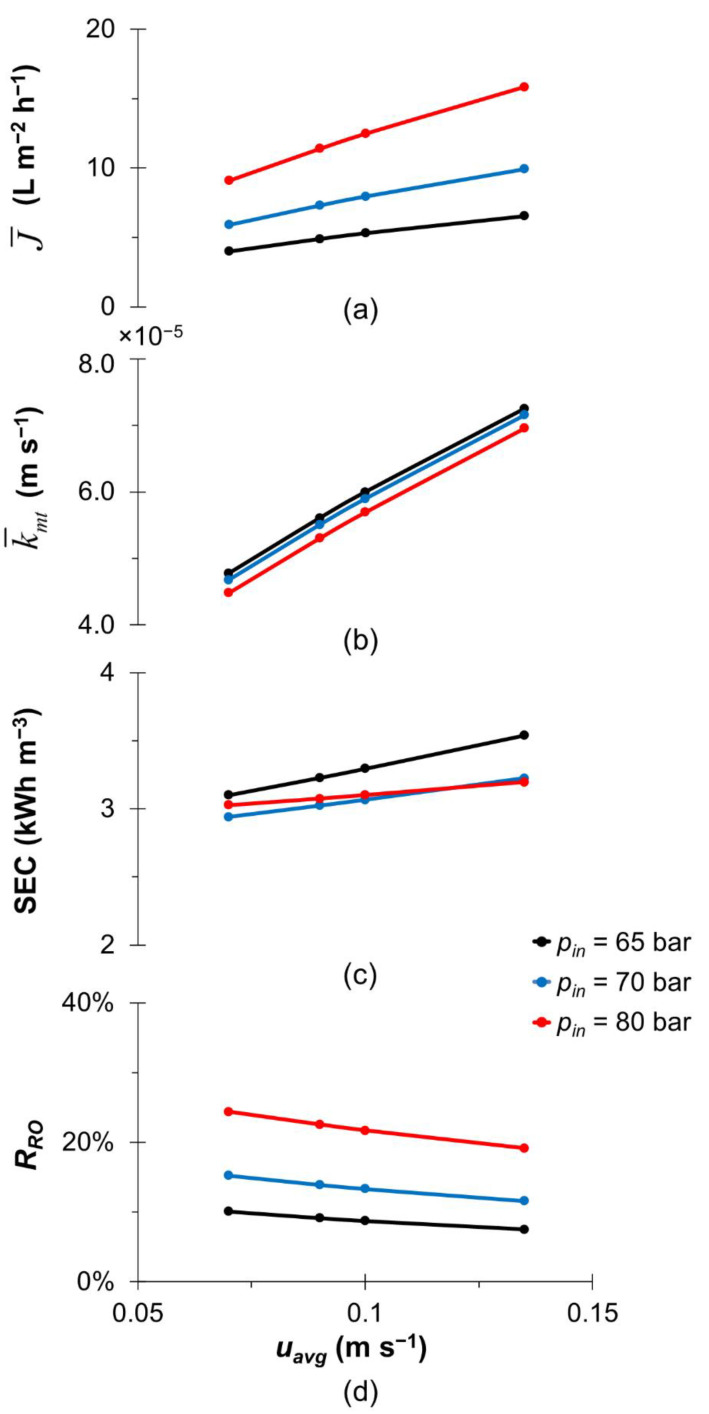
Effect of the inlet velocity (*u_avg_*) on the HBRO performance in terms of (**a**) the area-averaged flux $\left(\bar{J}\right)$, (**b**) area-averaged mass transfer coefficient $\left({\bar{k}}_{mt}\right)$, (**c**) specific energy consumption and (**d**) recovery ratio (*R_RO_*) for different inlet pressure (*p_in_*) at fixed *w_b_*_0,HB_ = 0.07, *η_R_* = 0.95 and *η_pump_* = 0.85.

**Figure 4 membranes-13-00248-f004:**
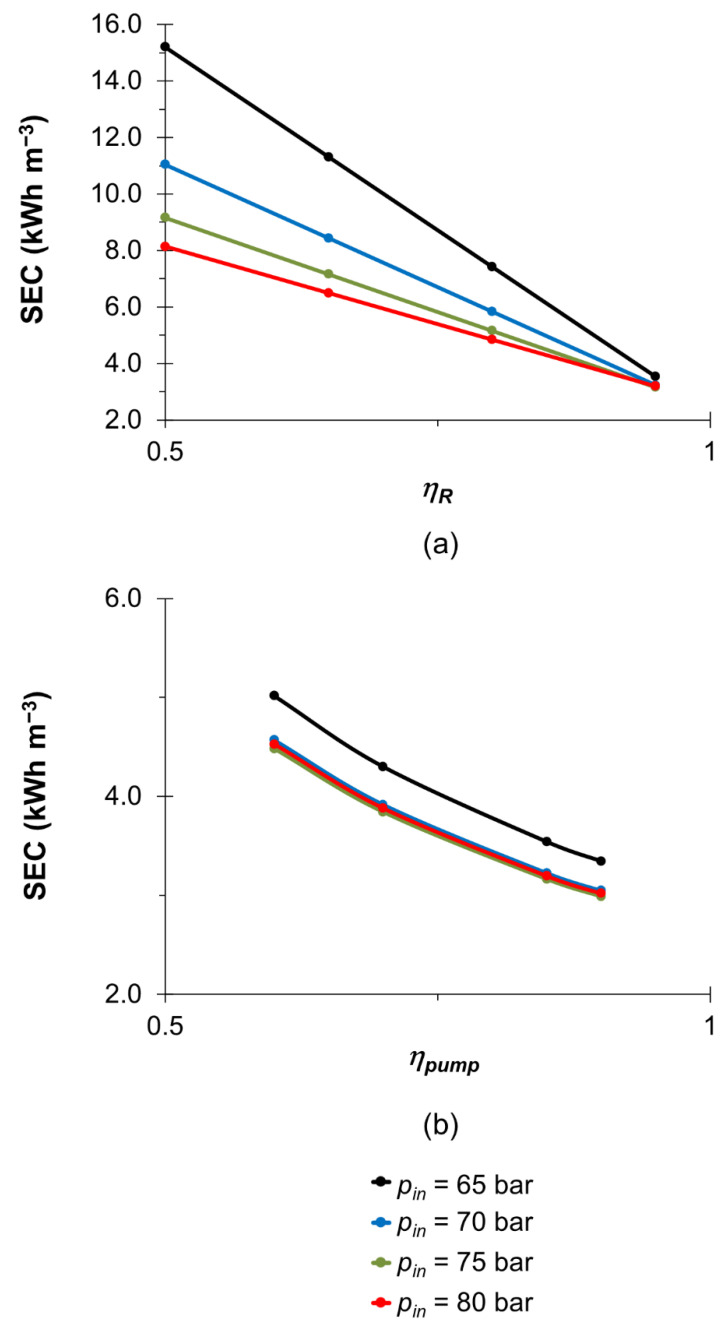
Effect of (**a**) retentate pressure recovery efficiency and (**b**) pump efficiency on SEC in the HBRO process at different inlet pressures (*p_in_*) with constant *w_b_*_0,HB_ = 0.07 and *u_avg_* = 0.135 m s^−1^.

**Table 1 membranes-13-00248-t001:** Case parameters used for SWRO and HBRO.

Case Parameters	SWRO	HBRO
Feed mass fraction, *w_b_*_0_	0.035	0.05–0.08
Inlet pressure, *p_in_* (bar)	65–80	
Inlet velocity, *u_avg_* (m s^−1^)	0.07–0.135	

**Table 2 membranes-13-00248-t002:** SWM module specifications and base conditions used for the case studies of SWRO and HBRO.

Parameters	Value
Membrane area of module, *A_m_* (m^2^)	28
Number of envelopes, *N*	14
Module length, *L_m_* (m)	1
Number of module units per pressure vessel	4
Channel width of module, *δ_ch_* (m)	1
Channel height, *h_ch_* (m)	0.001
Schmidt number, *Sc*	600
Membrane permeance, *L_p_* (m Pa^−1^ s^−1^)	6 × 10^−12^
Reflection coefficient, *σ*	1
Osmotic pressure coefficient, *φ* (Pa)	8.051 × 10^7^
Intrinsic rejection, *R*	0.996
Inlet velocity, *u_avg_* (m s^−1^)	0.07–0.135
Retentate pressure recovery efficiency, *η_R_*	0.95
Pump efficiency, *η_pump_*	0.85

## Data Availability

Data sharing not applicable.

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
