# Peer review of "Performance of Hypersaline Brine Desalination Using Spiral Wound Membrane: A Parametric Study"

_membranes, 2023, doi:10.3390/membranes13020248_

Round 1

Reviewer 1 Report (Previous Reviewer 1)

I am glad to see that the authors have revised the paper properly. Some minor issues/suggestions are as follows.

1) The results part seems a little stuffless (only 4 figures and 2 sub-sections 3.1 & 3.2) for a research paper. Can the authors add more research results?

2) Comment 3: there is a suggested reference missed. (Desalination 523 (2022) 115447)

3) Provide suggestions for improving the efficiency of hypersaline brine desalination.

Author Response

Reviewer 2 Report (New Reviewer)

The manuscript membranes-2201255 performed a parametric study of the performance of hypersaline brine desalination using spiral wound membrane, which is very useful for emerging technology.  The manuscript could be published in Membranes after making the following two points.

1. The range of the pressure was set to be 65-80 Bar. The upper bound of the pressure could be extended to a higher value. 

2. The temperature is an important parameter of the design and operation of RO system.  The influence of the temperature should be investigated.

Author Response

Reviewer 3 Report (New Reviewer)

Please see the file attached.

Round 2

Reviewer 1 Report (Previous Reviewer 1)

No further comments. A small problem:

Line 107&195: "Error! Reference source not found". What's this?

Author Response

Reviewer 1

Line 107&195: "Error! Reference source not found". What's this?

Response: We apologise for the confusion. These have been corrected on page 3 and 5, respectively.

Reviewer 2 Report (New Reviewer)

The influence of the temperature should be investigated. In ref.15, the hypersaline brine desalination was not included. A parametric study of  temperature effect should be addressed before being published on Membranes.  

Author Response

Reviewer 2

The influence of the temperature should be investigated. In ref.15, the hypersaline brine desalination was not included. A parametric study of temperature effect should be addressed before being published on Membranes. 

Response: We thank the author for the suggestion. We would like to clarify that the impact of temperature for low and high salinity waters has been reported from the literature. Therefore, this analysis is not carried out in this work. However, we have added “However, the positive effects of increasing feed temperature were found to be more significant for low salinity feed water, compared with high salinity water.” on page 2. Reference is added to support this statement.

Reviewer 3 Report (New Reviewer)

Dear authors,

Please send me a clean version of your paper and highlight all your changes made to evaluate your respond against the whole raised comments

Best wishes 

Author Response

Reviewer 3

Dear authors,

Please send me a clean version of your paper and highlight all your changes made to evaluate your respond against the whole raised comments.

Best wishes

Response: We apologise that the track changes have caused difficulty in reading the pdf. Therefore, we have highlighted the changes in blue colour so that the changes are more visible.

Round 3

Reviewer 3 Report (New Reviewer)

(ACCEPT)

The paper is perfect as the authors committed to resolve the raised comments and therefore the paper can be sent to the publication stream.

Author Response

We thank the reviewer for the acceptance.

This manuscript is a resubmission of an earlier submission. The following is a list of the peer review reports and author responses from that submission.

Round 1

Reviewer 1 Report

This study concentrates on the performance of hypersaline brine desalination using SWM. It is a meaningful study. I will recommend it after the authors addressing the following concerns:

1. How to validate the calculation results? Are there any validation experiments performed?

2. Section 3 “Results and Discussion”: It seems that the authors only provide some results, but have little discussion on these results. This section is weak now. I hope the authors can add more in-depth discussions.

3. Fig. 3: At the same applied pressure, a higher velocity enhanced mass transfer and permeate flux, but only causes a marginal increase on SEC (<3%). Is it really reasonable? Do you mean a higher inlet velocity is always better? A recent study on SWM element (https://doi.org/10.1016/j.desal.2021.115447) shows that the inlet velocity can’t be too high due to the sharp increase of FCP, which will damage the SWM element. The authors should discuss this issue more deeply, and in addition to the abovementioned study, the following references may also contribute to the discussion.
https://doi.org/10.1016/j.watres.2016.10.012
https://doi.org/10.1016/j.desal.2020.114508
https://doi.org/10.1016/j.watres.2021.117146

4. Fig.4: It shows the higher the retentate pressure recovery efficiency and pump efficiency, the lower the SEC. How to improve these efficiencies? Is there any upper limit of these efficiencies?

5. The sensitivity analysis is a little shallow. I suggest 1) add significance test and provide p-value in Table3; 2) conduct DOE (eg. using RSM method) to find out the optimal combination of these 4 parameters to achieve the lowest SEC.

6. Line 45-50: The authors stated that the energy consumption is extremely high (52-70 kWh/m3) for the thermal-based method, but as is known that high-pressure-driven membrane technology (eg. RO) also cause a high energy consumption. The authors should provide the specific value of energy consumption of RO for comparison.

7. Line 51: The word “famous” is improper, it should be replaced with “common”.

8.Line 129-130: The sentence needs to be recast.

Reviewer 2 Report

It is odd that a paper on RO is produced but the equation for flux is not given. There is reference to a recent paper that is not open access and a paper from 50 years ago. Key equations must be given.

It is even odder that there is a claim that "The results found that the variation in feed brine concentration has little impact on SEC for hypersaline brine desalination" when the result for feed at 100 bar (and presumably a concentration close to standard RO) gives a value of 4 kWh per m3 whilst the result for a hypersaline brine is 10 kWh per m3 (see Figure 3c).

Given that there are 5 authors on this paper, does this mean that five people have OK'ed this counter-intuitive claim. 

By the way, a paper in Desalination from four years (Desalination 431, 151-156, 2018) suggested that performance at lab scale (flat sheet) is not replicated at pilot scale (spiral wound). Thus a big assumption is being made to assume that the CFD result can be used as a guide. Believe the values of mass transfer coefficient (Fig 2 c) are much larger than experimental values found in the literature. A check is essential.

Furthermore the compressibility of the membrane at high pressure and the decrease in performance that results is ignored (see work of Elimelech at Yale)